# Growth, Metabolite, Antioxidative Capacity, Transcriptome, and the Metabolome Response to Dietary Choline Chloride in Pacific White Shrimp *Litopenaeus vannamei*

**DOI:** 10.3390/ani10122246

**Published:** 2020-11-30

**Authors:** Maoxian Huang, Hongxing Lin, Chang Xu, Qiuran Yu, Xiaodan Wang, Jian G. Qin, Liqiao Chen, Fenglu Han, Erchao Li

**Affiliations:** 1Key Laboratory of Tropical Hydrobiology and Biotechnology of Hainan Province, Hainan Aquaculture Breeding Engineering Research Center, College of Marine Sciences, Hainan University, Haikou 570228, China; maoxian_huang@163.com (M.H.); linhongxing688@163.com (H.L.); cxu@hainanu.edu.cn (C.X.); yu974985801@126.com (Q.Y.); 2School of Life Sciences, East China Normal University, Shanghai 200241, China; xdwang@bio.ecnu.edu.cn (X.W.); lqchen@bio.ecnu.edu.cn (L.C.); 3School of Biological Sciences, Flinders University, Adelaide 5001, Australia; jian.qin@flinders.edu.au

**Keywords:** *Litopenaeus vannamei*, choline chloride, transcriptome, metabolome, antioxidant capacity

## Abstract

**Simple Summary:**

Choline is a vitamin-like nutrient and has many metabolic and physiological functions in aquatic animals. Unfortunately, the information on the optimal requirement of dietary choline in *Litopenaeus vannamei* is limited, and the molecular and metabolic mechanisms of choline on *L. vannamei* are unclear. Hence, in this study, the growth performance, whole-body composition, serum characteristics, hepatopancreatic antioxidant indexes, serum metabolome and hepatopancreas transcriptome were performed. In this study, the growth of *L. vannamei* was not affected by dietary choline. Dietary choline played an important role in arachidonic acid and glycerophospholipid metabolism and decreased the oxidant damage of *L. vannamei*, while excessive choline can inhibit the digestion of protein and reduce the whole-body crude protein in shrimp. Based on the results of weight gain and lipid peroxidation reduction, 1082 mg/kg dietary choline could meet the growth requirement of *L. vannamei*, but 2822 mg/kg dietary choline was needed to reduce peroxidation damage. The present study would provide valuable information on the requirement of choline in *L. vannamei*, and help to understand the molecular and metabolic mechanisms of choline in shrimp.

**Abstract:**

To determine the response of Pacific white shrimp *Litopenaeus vannamei* to different levels of dietary choline, juvenile white shrimp (1.75 ± 0.09 g) were fed six semi-purified diets supplemented with 0 (control), 2000, 4000, 6000, 8000, and 12,000 mg/kg choline chloride for eight weeks. Growth performance, whole-body composition, serum characteristics and hepatopancreatic antioxidant indexes were evaluated. Meanwhile, serum metabolome and hepatopancreas transcriptome were performed to examine the overall difference in metabolite and gene expression. The weight gain, survival, specific growth rate, condition factor and hepatosomatic index were not affected by dietary choline levels. The shrimp fed 6000 mg/kg dietary choline chloride gained the maximal whole-body crude protein, which was significantly higher than that of shrimp fed with 12,000 mg/kg dietary choline. Serum total cholesterol of shrimp fed 6000 mg/kg dietary choline was higher than that in shrimp fed 4000 mg/kg choline. Dietary choline significantly decreased malondialdehyde content, superoxide dismutase, and glutathione peroxidase activities in shrimp hepatopancreas. Compared with the shrimp fed 6000 mg/kg dietary choline chloride, the glycerophospholipid metabolism pathway was significantly enriched in the shrimp fed 0 mg/kg dietary choline chloride, and the choline content and bile salt-activated lipase-like expression were upregulated. The expression of trypsin-1-like in protein digestion and absorption pathway was significantly downregulated in the shrimp fed 12,000 mg/kg dietary choline chloride. Apolipoprotein D might be a potential biomarker in shrimp, and dietary choline played an important role in lipid metabolism, especially in the reduction of oxidative damage in *L. vannamei*. Based on the results of weight gain and degree of oxidative damage, 1082 mg/kg dietary choline could meet the growth requirement of *L. vannamei*, but 2822 mg/kg dietary choline was needed to reduce peroxidation damage.

## 1. Introduction

Choline is a vitamin-like nutrient and is essential to the growth of some aquatic animals [1,2], and choline chloride is widely used as a source of dietary choline to determine the optimal level of dietary choline for various aquatic animals, such as *Sciaenops ocellatus* [1], hybrid tilapia (*Oreochromis niloticus* × *O. aureus*) [3], *Penaeus monodon* [2], and *Marsupenaeus japonicus* [4], *Cyprinus carpio* var. Jian [5]. Unfortunately, the information on the optimal requirement of dietary choline in crustacean species is fragmental and inconsistent, and the discrepancy may be related to animal species, growth stage, diet ingredient, culture condition, and evaluation criteria [2,4]. For example, the growth and survival of *M. japonicus* (about 0.004 and 0.01 g initial body weight, respectively) were reportedly improved by feeding on 600 and 1200 mg/kg dietary choline [6,7]. However, Deshimaru and Kuroki (1979) had concluded that dietary choline chloride did not affect the growth of *M. japonicus* with 0.5 g initial body weight [8]. In juvenile *P. monodon*, the optimum dietary choline requirement is affected by dietary lipid concentration [9]. Thus, there is a need to accurately determine the choline requirement of the same species.

Dietary choline plays many fundamental functions as a structural component of phosphatidylcholine and acetylcholine, a precursor of the methionine and betaine, and a metabolic methyl donor [7,9,10,11]. Choline also has many metabolic and physiological functions in aquatic animals. Total lipids in the liver and plasma significantly decreased in *S. ocellatus* fed a choline-free diet [1]. In contrast, the hepatosomatic lipid concentration and hepatopancreatic index are higher in choline-deficient *P. monodon* [9] and *Epinephelus lanceolatus* [12]. The whole-body crude protein is also higher in *P. monodon* fed given 4000 and 7000 mg/kg dietary choline [2]. Several studies have revealed that choline can significantly affect serum biochemical contents such as triglyceride and cholesterol in aquatic animals [1,3,12]. Moreover, dietary choline can regulate the antioxidant system to decrease oxidative damage in juvenile Jian carp [5] and Nile tilapia [13]. However, these studies have only focused on choline deficiency but ignored the effect of choline overdose on organism performance, and the molecular mechanisms of these physiological effects are not clear.

The Pacific white shrimp *Litopenaeus vannamei* is one of the major penaeid species in aquaculture worldwide [14]. The choline chloride requirement of *L. vannamei* has been estimated without supplemental phospholipid or cholesterol, and the study is limited to the analyses of moisture and lipid content in shrimp [15]. Dietary cholesterol is needed for optimal growth of *L. vannamei* [16], and the cholesterol-deficient diet can reduce shrimp growth and indirectly affect the optimal choline requirement of *L. vannamei*. This study aims to compare the growth performance, whole-body proximate composition, serum metabolites, antioxidant, and digestive enzyme activities in the hepatopancreas of *L. vannamei* fed a casein-based diet enriched with different levels of choline chloride. Based on metabolome and transcriptome analyses, we identified the key genes, metabolites, and pathways that are related to the molecular and metabolic mechanisms between shrimp fed choline-deficient and overdosed diets.

## 2. Materials and Methods

### 2.1. Experimental Diets

The isonitrogenous (about 42.20% crude protein) and isolipidic (about 7.99% crude lipid) semi-purified diets were formulated with six graded levels of choline chloride (0, 2000, 4000, 6000, 8000, and 12,000 mg/kg diet). Casein was the main protein source in the test diets. Fish oil, soybean oil, lecithin and cholesterol were the source of lipids. The experimental diets were formulated with a choline-free vitamin premix. A mixture of amino acids including betaine, L-glutamic, L-alanine acid, and glycine (Shanghai Sangon Biotech Co., Ltd., Shanghai, China) was used as an attractant [2]. Butylated hydroxytoluene (BHT), an effective antioxidant, was added at a concentration of 0.005% to prevent lipid oxidation in the diets. After ground and sieved through a 60-mesh sieve, all dry ingredients were mixed thoroughly before oil was added. The choline chloride (reagent grade > 98%; Sangon Biotech, Shanghai, China) was supplemented to the basal diet while reducing the same amount of α-cellulose. Finally, about 300 mL/kg distilled water was added to the mixture to produce a stiff dough, which was wet-extruded into 2.5-mm diameter pellets at an extrusion temperature of less than 50 °C (CD4-1TS extruder, SCUT industrial factory, Guangzhou, China). The pellets were dried by blowing air at room temperature until reaching <10% moisture. Pellets were sieved by a 40-mesh sieve and stored at −20 °C. Ingredients and proximate composition of the basal diet are given in Table 1. The animal ethics protocol was approved by the Hainan University Experimental Animal Ethics Committee (No. HNUAUCC-2020-00005).

The total choline in diets were detected by the Guangdong Institute of Analysis (Guangzhou, China) according to the national standards of China (GB/T 14924. 11-2001/3.7) using spectrophotometry. Briefly, the total choline in the diets was extracted by the alkali treatment process and purified by Florisil column chromatography, and then it reacted with Reinecke salt to form a pink choline—Reinecke salt complex. After eluting with acetone, this complex has an absorption maximum at 526 nm, and its absorption is proportional to the concentration of choline. The detection limit of choline was 0.1 mg, and the absolute value of the relative deviation of repeated detection results was ≤10%. The actual levels of total choline in the diets were 1082, 2822, 4770, 6450, 9120 and 13,100 mg/kg, respectively. Total choline is the sum of multiple choline compounds, and the choline detected in the basal diet may be choline, phosphatidylcholine, glycerophosphocholine, phosphocholine, and sphingomyelin in ingredients [17,18].

### 2.2. Growth Trial and Sampling

Juvenile *L. vannamei* were obtained from a shrimp hatchery in Wenchang, China. During a one-month acclimation process, the animals were maintained in an indoor cement pool and fed with a commercial diet (42% crude protein, 8% crude lipid) three times a day (0700, 1200, and 1800), and about three-quarters of the seawater in the pool was renewed once a day. A total of 720 healthy shrimp (1.75 ± 0.09 g) were randomly assigned to 24 cylindrical fiber-glass tanks filled with 100 L sand-filtered seawater. Each treatment was randomly assigned to four replicated tank with 30 shrimp each. Shrimp were fed three times a day at 0700, 1200, and 1800 with a daily ration of 5% body mass for eight weeks. When approximately 60% of seawater in tanks were daily exchanged, meantime the feces, uneaten feed, and exuviae were removed. The dead shrimp were removed, weighed, and recorded immediately. The natural illumination condition was used during the feeding trial and water quality was kept at salinity 30–33‰, temperature 28–30 °C, pH 7.8–8.2, dissolved oxygen 4.8–6.4 mg/L and ammonia-N < 0.05 mg/L.

All shrimp in each tank were fasted for 24 h and then anesthetized in an ice slurry. Shrimp were counted, weighed (0.01 g) and measured the total body length (0.1 cm) to calculate survival (%), condition factor (%), weight gain (%), and specific growth rate (% day^−1^). Three whole shrimp per tank were randomly selected and kept at −20 °C for the analysis of whole-body composition. Hemolymph samples were extracted from the pericardial cavity of remaining shrimp in each tank using 1 mL disposable sterilized syringes, stored in 1.5 mL Eppendorf tubes at 4 °C for 24 h, and then centrifuged at 4 °C, 5000 r/min for 10 min (3–18KS, Sigma, Osterode am Harz, Germany). The serum samples were stored in 200 μL Eppendorf tubes at −80 °C until biochemical analysis and metabolomic analysis. The hepatopancreas samples were taken from the aforesaid processed shrimp to calculate the hepatosomatic index (%) and were stored at −80 °C until subsequent analysis.

Survival, weight gain, specific growth rate, condition factor, and hepatosomatic index were evaluated using the following formulae:(1)Survival (%) = 100% × final shrimp number/initial shrimp number.(2)Weight gain (%) = 100% × (final weight (g) − initial weight (g))/initial weight (g).(3)Specific growth rate (% day^−1^) = 100% × (ln (final weight) − ln (initial weight))/days.(4)Condition factor (%) = 100% × final weight (g)/(body length (cm))^3^.(5)Hepatosomatic index (%) = 100% × wet hepatopancreas weight (g)/wet body weight (g).

### 2.3. Proximate Composition Analysis

The proximate composition of experimental diets and shrimp whole body were analyzed following the standard methods of the Association of Official Analytical Chemists [19]. Briefly, the moisture of diets and whole body were determined by drying to a constant weight at 105 °C (WFO-520, EYELA, Tokyo, Japan). The crude protein contents were measured by using the Dumas combustion method (Elementar rapid N exceed, Frankfurt, Germany). Crude lipid contents were determined via petroleum ether extraction using a Soxhlet extractor. The ash contents were determined using a muffle furnace (SX2-4-10N, Yiheng, Shanghai, China) at 550 °C for 8 h.

### 2.4. Biochemical Assay

The contents of total protein, total cholesterol, glucose, and triacylglycerol in serum samples were determined in eight replicates per diet. Eight hepatopancreases per treatment were weighed and homogenized in the pre-chilled 0.86% saline solution (1:10, *w*/*v*) at a frequency of 60 Hz at 4 °C for 30 s (Tissuelyser-24, Shanghai Jingxin Technology, Shanghai, China), centrifuged at 4 °C with 1500× *g* for 15 min (3-18KS, Sigma, Germany), and the supernatant was collected to measure total protein content, malondialdehyde (MDA) content, the activities of superoxide dismutase (SOD), glutathione peroxidase (GSH-Px), α-amylase (AMS) and lipase (LPS). All the biochemical parameters were determined according to the manufacturer’s instructions using commercial assay kits (Nanjing Jiancheng Bioengineering Institute, Nanjing, China).

### 2.5. Untargeted Metabolomic Analysis

#### 2.5.1. Sample Preparation

A total of 18 serum samples from shrimp fed 0, 6000, and 12,000 mg/kg choline chloride were obtained for the metabolomic analysis, and six biological replicates were used per group. Firstly, 400 µL of methanol-acetonitrile (1:1, *v*/*v*) was mixed with a serum sample (100 µL) and subjected to ultrasound extraction for 30 min (5 °C, 40 kHz). Then the solution was incubated at −20 °C for 30 min and centrifuged at 4 °C with 13,000× *g* for 15 min. Subsequently, the supernatant was dried under N_2_ (JXDC-20, Shanghai Jingxin Technology, Shanghai, China) and re-dissolved in 100 µL of acetonitrile-water (1:1, *v*/*v*). The re-dissolvent was ultrasonically extracted at 5 °C for 5 min and centrifuged for 5 min at 4 °C with 13,000× *g*. Finally, the supernatant was added to the sample bottle for detection. A mixture of equal quantities (20 µL per sample) was extracted from all samples and used as a quality control (QC) sample for liquid chromatography-mass spectrometry (LC-MS) analysis.

#### 2.5.2. LC-MS Analysis

The LC-MS-based serum metabolic profiling adopted a Vanquish™ Horizon UHPLC system (Thermo Scientific, Germering, Germany) coupled with a Thermo Scientific™ Q Exactive™ Plus Hybrid Quadrupole-Orbitrap™ mass spectrometer. The parameters of chromatography were as follows: column: Ethylene Bridged Hybrid C18 (100 mm × 2.1 mm, 1.7 μm, Waters, Milford, Massachusetts, USA); gradient mobile phase: (A) deionized water containing 0.1% formic acid, (B) acetonitrile/isopropanol (1:1, *v*/*v*) mixture containing 0.1% formic acid; flow rate: 0.4 mL/min; sample injection volume: 2 μL; column temperature: 40 °C. The mobile phase gradient was: 0–3 min, A: 95–80%; 3–9 min, A: 80–5%; 9–13 min, A: 5–5%; 13–13.1 min, A: 5–95%; 13.1–16 min, A: 95–95%. The conditions of MS included the scan ranges (M/Z): 70–1050; Sheath gas flow rate (psi): 40; Aus gas flow rate (psi): 10; Aus gas heater temp (°C): 400; Normalized collision energy (V): 20–40–60; and IonSpray Voltage Floating (V): positive mode (ESI^+^), +3500; negative mode (ESI^−^), −2800. A QC sample was inserted for every 6 analytical samples during the experiment to evaluate the stability of the analytical system and assess the reliability of the results.

#### 2.5.3. Metabolomic Data Analysis

The raw data obtained from the LC-MS analysis of all samples were processed initially using Progenesis QI software (Waters Corporation, Milford, MA, USA). comment HMDB database (http://www.hmdb.ca/), METLIN database (https://metlin.scripps.edu/), KEGG database (https://www.genome.jp/kegg/), and a self-built database were selected for retrieval. Multivariate analyses including principal component analysis (PCA) and orthogonal partial least squares discrimination analysis (OPLS-DA) were performed by using ROPLS software (v1.6.2). Also, the OPLS-DA models were validated using a permutation test with 200 as the permutation number. To select differential metabolites and potential biomarkers, the variable importance in the projection (VIP, VIP > 1) values of metabolites in the OPLS-DA model and *p* values (*p* < 0.05) acquired from the *t*-test analysis were regarded as the screening condition.

### 2.6. Transcriptomics Analysis

The total RNA of hepatopancreas samples dissected from individual shrimp fed 0, 6000, and 12,000 mg/kg choline chloride (3 shrimps per diet) was extracted using the TRIzol kit (Invitrogen, Carlsbad, CA, USA). The integrity, quality, and quantity of RNA were examined with an Agilent 2100 Bioanalyzer (Agilent Technologies, Santa Clara, CA, USA) and a NanoDrop 2000 spectrophotometer (Thermo Scientific, Delaware, USA). Nine cDNA libraries were constructed using a TruSeq^TM^ RNA sample prep kit (Illumina, San Diego, CA, USA) according to the manufacturer’s procedure. Nine libraries were sequenced using Illumina Novaseq 6000 sequencing platform (2 × 150 bp read length). The raw data were filtered with SeqPrep and Sickle software to obtain clean data. Then clean reads were mapped to the *L. vannamei* reference genome by using HISAT2 software and were assembled using Stringtie (v1.3.3b). The assembled transcriptome was annotated using the NR, Pfam, Swiss-Prot, COG, GO, and KEGG databases. Gene expression levels were calculated using RSEM software (v1.3.1) with the transcripts per million (TPM) method. Differentially expressed genes (DEGs) were analyzed using the DESeq2 software (v1.24.0), and the *p*-values were adjusted using the Benjamini-Hochberg method (*P* adjust). The selection criteria were *p* adjust < 0.05 and |log_2_ (fold change, FC)| ≥ 1.

### 2.7. Statistical Analysis

Statistical analysis was performed using SPSS 17.0 for Windows (SPSS Inc., New York, NY, USA). The outliers in each treatment were removed using box plot analysis. All the filtered data were first tested to confirm normal distribution and homogeneity of variance. One-way analysis of variance (ANOVA) was used to test the main effect of dietary manipulation using Tukey’s honestly significant difference (HSD) test as post hoc test [20]. If both the raw data and log-transformed data showed a lot of heteroscedasticity, the analysis was performed using Welch’s ANOVA and Games-Howell post hoc test [20]. All data were expressed as mean ± standard error (SE), and statistical significance was set at *p* < 0.05.

## 3. Results

### 3.1. Growth Performance and Whole-Body Proximate Composition

No significant differences were found in shrimp survival, weight gain, specific growth rate, condition factor, and hepatosomatic index after eight-week cultivation (Table 2). The crude protein of shrimp fed 12,000 mg/kg dietary choline chloride was significantly lower than that in shrimp fed 6000 mg/kg dietary choline chloride (Table 3). The moisture, ash, and crude lipid contents of shrimp were not affected by the dietary treatments (Table 3).

### 3.2. Serum Metabolites and Hepatopancreatic Digestive Enzymes

In serum samples, the total cholesterol content in shrimp supplemented with 6000 mg/kg dietary choline was significantly higher than those of shrimp received 4000 mg/kg dietary choline (Table 4). The total protein, glucose, and triglyceride contents were not affected by the dietary treatments (Table 4). Both α-amylase and lipase activities of shrimp were not affected by the dietary treatments (Table 5).

### 3.3. MDA and Antioxidant Parameters

In the hepatopancreas, the shrimp in the control gained significantly higher MDA contents and SOD activities than those of shrimp fed other diets (*p* < 0.05). The activity of GSH-Px was the highest in shrimp given the control diet, and significantly higher than that in shrimp given 4000 mg/kg dietary choline chloride (Table 5).

### 3.4. Metabolome Analysis

The PCA scores plots showed great clusters of the quality control samples in both positive and negative ion modes and confirmed the reliable repeatability and precision of the data in this study (Figure 1A,B). The OPLS-DA score plots showed great clear discriminations in the three compared groups, indicating that different dietary choline levels altered serum metabolomics profiles of *L. vannamei* (Figure 1C–H).

A total of 42, 23 and 45 differential metabolites were identified between the pair comparisons of 0 versus 6000, 6000 versus12,000, and 0 versus 12,000, respectively. The relative expression levels and VIP values of differential metabolites were showed in Figure 2. Venn diagram was constructed to identify the shared and unique differential metabolites in the three compared groups (Figure 3A). The HMDB compound classification analysis showed that the amino acids, peptides, and analogues, eicosanoidsand, glycerophosphoethanolamines, and glycerophosphocholines were the main differential metabolite categories in the three compared groups (Figure 4A,C,E). According to the KEGG enrichment analysis, arachidonic acid metabolism and glycerophospholipid metabolism pathways were significantly enriched in 0 versus 6000 and 0 versus 12,000 compared groups (*p* < 0.05, Figure 4B,D). The contents of choline, 5-trans-PGE_2_, and 15-deoxy-delta-12,14-PGJ_2_ were decreased in shrimp fed 6000 and 12,000 mg/kg dietary choline. The L-proline in the protein digestion and absorption pathway was upregulated in shrimp given 0 and 12,000 mg/kg dietary choline. The details of differential metabolites were listed in the Appendix A.

### 3.5. Transcriptome Analysis

After quality filtering, 48,699,065, 51,050,008 and 51,946,736 clean reads (mean value, n = 3) were obtained from the shrimp fed with 0, 6000 and 12,000 mg/kg dietary choline, respectively. The GC content of clean data accounted for 50.54–50.93%. The percentage of Q30 bases (Phred score of 30) was higher than 95.67% and the error rate was all <0.03%, indicating the high quality of reads. The average of 88.12–88.70% of the clean reads was mapped to the reference genome (Table 6). A total of 16,044 (85.03% of the total (18,868)) expressed genes were annotated in six databases (Figure 5D), and the common and unique DEGs in the three compared groups were identified by the Venn diagram (Figure 3B). A total of 88, 52, and 144 DEGs were identified in 0 versus 6000, 0 versus 12,000, and 6000 versus 12,000 compared groups, respectively. Pathways belonging to human diseases were ignored in this study. The protein digestion and absorption pathway were significantly enriched in 0 versus 6000 and 6000 versus 12,000 compared groups (Figure 5A,C). Moreover, the gene expression of carboxypeptidase A1-like was upregulated in shrimp given 0 and 12,000 mg/kg choline chloride, while trypsin-1-like was only downregulated in shrimp given 12,000 mg/kg choline chloride. Gene expressions of apolipoprotein D-like, bile salt-activated lipase-like, diacylglycerol kinase epsilon-like, low-density lipoprotein receptor 1-like, and glutathione S-transferase 1-like were upregulated in the shrimp given 0 mg/kg dietary choline chloride. The DEGs without clear gene descriptions were eliminated and the details of the remaining DEGs were listed in Appendix A.

## 4. Discussion

Choline has been considered as an essential vitamin for some fish species and other animals [2,5]; in this study, no significant differences in weight gain, survival, hepatosomatic index, and condition factor were found between different groups. It indicated that the total choline level of 1082 mg/kg in the basal diet was sufficient for the normal growth of the *L. vannamei*. A previous study showed that the dietary choline requirement of *L. vannamei* (0.72 ± 0.20 g) based on the instantaneous growth rate is 871 mg/kg, but choline requirement is not evident when lecithin is more than 1.5% of the diet [15]. However, 3254.1 and 6488.3 mg/kg dietary choline were estimated to be the optimal choline requirements for juvenile *L. vannamei* (initial weight of 0.30 ± 0.00 g) based on percent weight gain [21]. The discrepancy of optimal choline requirements may be related to animal growth stage, diet ingredient, culture condition, and evaluation criteria [2,4]. Furthermore, it is generally believed that there may be interactions between dietary choline, betaine, methionine, and phosphatidylcholine, and these ingredients may influence the requirement for choline [22]. Thus, more studies are needed in the future to determine the optimal choline requirement of *L. vannamei*.

Trypsin as an important digestive protease mainly participates in food digestion, hydrolysis, and activation of zymogens [23]. Compared with shrimp given 6000 mg/kg dietary choline chloride, the expression of trypsin-1-like was downregulated in the protein digestion and absorption pathway of shrimp given 12,000 mg/kg dietary choline. Thus, 12,000 mg/kg dietary choline might inhibit trypsin secretion, damages the protein utilization and deposition of *L. vannamei*, thereby reducing the whole-body crude protein content. Paradoxically, the expressions of carboxypeptidase A1-like were upregulated in shrimp given 0 and 12,000 mg/kg dietary choline chloride. Carboxypeptidase A is a digestive carboxypeptidase and serves in the degradation of proteins in the digestive tract [23]. Thus, the mechanisms of how dietary choline chloride regulated the whole-body crude protein of *L. vannamei* need further study.

Overproduced reactive oxygen species (ROS) would lead to cell and tissue oxidative damage [24]. Generally, MDA is a key metabolite production and a good biomarker of lipid peroxidation [25]. This study showed that the MDA contents in shrimp hepatopancreas were decreased by dietary choline chloride, suggesting that choline supplement can decrease oxidative damage in the hepatopancreas of *L. vannamei*. Similarly, higher MDA contents were found in the kidney, hepatopancreas and intestine of Jian carp fed the choline-deficient diet [5,26]. Choline deficiency induced rats fatty liver and lower levels of antioxidants, resulting in lipid peroxidation [27]. In this study, the higher SOD and GSH-Px activities in choline-deficient shrimp may be related to the high ROS levels. ROS are rapidly eliminated by the antioxidant defense system [28]. The SOD converts superoxide anion into hydrogen peroxide that passes freely through membranes and GSH-Px plays a vital role in removing hydrogen peroxide from cells [29]. Furthermore, compared with the 12,000 mg/kg dietary choline chloride group, transcriptome results showed that glutathione S-transferase 1-like gene expression was significantly upregulated in the shrimp given 0 mg/kg dietary choline chloride. Glutathione S-transferases protect aquatic organisms from oxidative damage [30]. However, scanty evidence is available concerning the relationship between choline and antioxidant response in *L. vannamei*. As reported, choline may give a rise to antioxidant activity as decomposers of hydroperoxides [31].

In *S. ocellatus*, dietary choline induced higher concentrations of cholesterol, cholesterol esters, triglycerides, and phosphatidylcholine in plasma [1]. Blood triglyceride, cholesterol, and phospholipid concentrations were higher in hybrid tilapia fed the diets supplemented with choline [3]. Choline acts as a lipotropic factor preventing excessive lipid accumulation [32]. In fish, choline deficiency can impair hepatic lipoprotein secretion, resulting in excessive accumulation of lipids in the liver [12]. The hepatic secretion of lipoprotein requires active synthesis of phosphatidylcholine [33]. Choline is a major component of phosphatidylcholine and sufficient phosphatidylcholine promotes lipid transport [9,12]. As a result, the contents of lipid metabolites in the blood increase when fish are supplied with dietary choline. Similarly, *P. monodon* fed choline-deficient diets have higher hepatopancreatic lipid concentration than that of shrimp fed with supplemented choline chloride [9]. On the contrary, higher concentrations of lipid metabolites in hemolymph, including high-density lipoprotein and low-density lipoprotein, were found in *L. vannamei* fed the basal diet [21]. In this study, compared with shrimp fed 0 mg/kg choline, phosphatidylcholine (lecithin) was downregulated in shrimp fed 6000 mg/kg choline (Appendix A). Therefore, choline might reduce phosphatidylcholine content in *L. vannamei*, resulting in a decrease in hemolymph lipid content. Moreover, dietary choline had no statistical effect on hemolymph cholesterol and triglyceride contents of *L. vannamei* [21]. In this study, no significant difference was found in serum triglyceride among all treatment groups, while the serum total cholesterol in shrimp fed 6000 mg/kg choline was significantly higher than that in shrimp fed 4000 mg/kg choline. Interestingly, no significant difference was found though the serum total cholesterol content of shrimp given 12,000 mg/kg choline was about 1.8 times higher than that of shrimp given 0 mg/kg choline. In hamsters and rabbits, plasma cholesterol levels are low in part by stimulating the production of mRNA for the low-density lipoprotein receptor in the liver [34]. In this study, the gene expression of low-density lipoprotein receptor 1-like was upregulated in shrimp fed the basal diet. Thus, we postulate that dietary choline can increase serum cholesterol by impairing gene expression of low-density lipoprotein receptor.

Compared with shrimp given the basal diet, significantly lower serum choline was gained in shrimp fed 6000 mg/kg choline chloride (Appendix A). Moreover, the results of the transcriptome showed that the expression of bile salt-activated lipase-like (EC: 3.1.1.7) was down-regulated, so the synthesis of choline from O-Acetylcholine was inhibited in shrimp fed 6000 mg/kg choline (Appendix A). A possible explanation is that the decreased serum choline could help maintain the homeostasis of hemolymph osmotic pressure when cholesterol content increased, because choline could be metabolized to organic osmolytes such as phosphatidylcholine and betaine [35].

Apolipoprotein D plays various physiological roles in an organism [36,37]. The overexpression of apolipoprotein D can protect *Drosophila* against acute oxidative stress [37]. The loss of mouse apolipoprotein D function can increase the sensitivity to oxidative stress and aggravate the level of brain lipid peroxidation [38]. Particularly, arachidonic acid, a precursor for prostaglandin synthesis, is known to be mobilized from membranes upon oxidative stress, and apolipoprotein D can bind and mobilize arachidonic acid specifically for the synthesis of prostaglandin [36]. Moreover, this binding may quench deleterious molecules such as lipid peroxides or scavenge other free radicals [37]. In this study, the 5-trans-PGE_2_ and 15-deoxy-delta-12,14-PGJ_2_ in the arachidonic acid metabolism pathway increased in shrimp fed 0 mg/kg dietary choline. A previous study has shown that the 15-deoxy-delta-12,14-PGJ_2_ reduced the cytotoxicity caused by hydrogen peroxide [39]. Another vital oxidative stress-preventing role of 15-deoxy-delta-12,14-PGJ_2_ is its ability to upregulate the expression of glutamylcysteine synthetase, a rate-limiting enzyme in glutathione synthesis, and induces the synthesis of glutathione [40]. Therefore, the higher expression of apolipoprotein D and 15-deoxy-delta-12,14-PGJ_2_ are the possible physiological adaptation for choline-deficient shrimp to lipid peroxidation. With the data available so far, it is difficult to clarify the physiological role of other prostaglandins such as 11-deoxy-PGE_1_, 5-trans-PGE_2_, 13,14-Dihydro-15-keto-PGE_2,_ and Ent-Prostaglandin F_2α_ in aquatic animals. Prostaglandins play a vital role in regulating many normal cellular functions [41]. The synthesis of phosphatidylcholine also depends on prostaglandins, which modulates biosynthetic enzymes [42]. The PGE_2_ and PGF_2α_ can stimulate the formation of choline in a dose-dependent manner [43,44].

## 5. Conclusions

The growth of *L. vannamei* was not significantly improved by dietary choline chloride. The dietary choline decreased the oxidant damage of *L. vannamei*, while excessive choline can inhibit the digestion of protein, reduce protein deposition, and reduce the whole-body crude protein in shrimp. In this study, the metabolome and transcriptome were used to analyze the molecular mechanism of choline on *L. vannamei*. Based on the results of weight gain and lipid peroxidation reduction, 1082 mg/kg dietary total choline could meet the normal growth of *L. vannamei*, but 2822 mg/kg total choline could reduce peroxidation damage in hepatopancreas.

## Figures and Tables

**Figure 1 animals-10-02246-f001:**
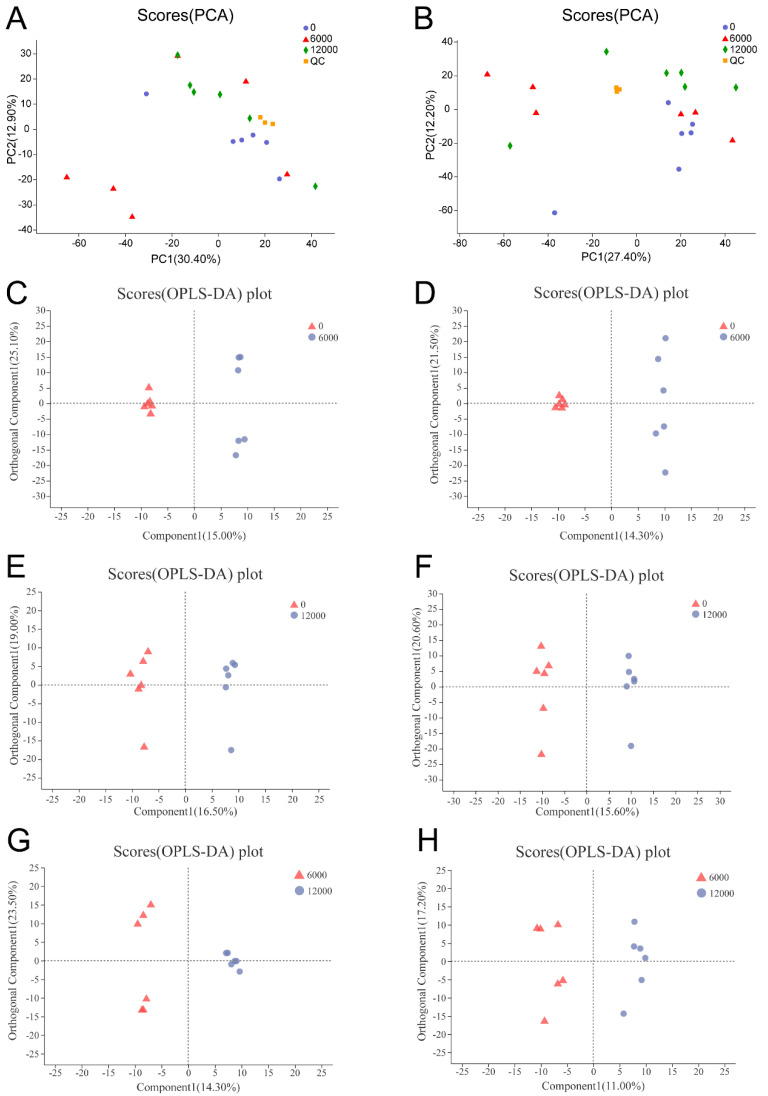
PCA scores plot of *L. vannamei* given 0, 6000 and 12,000 mg/kg dietary choline chloride: (**A**) ESI^+^ mode; (**B**) ESI^−^ mode. QC is the abbreviation of quality control sample. OPLS-DA scores plot of *L. vannamei* in the three compared groups: (**C**) ESI^+^ and (**D**) ESI^−^ modes of shrimp given 0 versus 6000 mg/kg choline; (**E**) ESI^+^ and (**F**) ESI^−^ modes of shrimp given 0 versus 12,000 mg/kg choline; (**G**) ESI^+^ and (**H**) ESI^−^ modes of shrimp given 6000 versus 12,000 mg/kg choline.

**Figure 2 animals-10-02246-f002:**
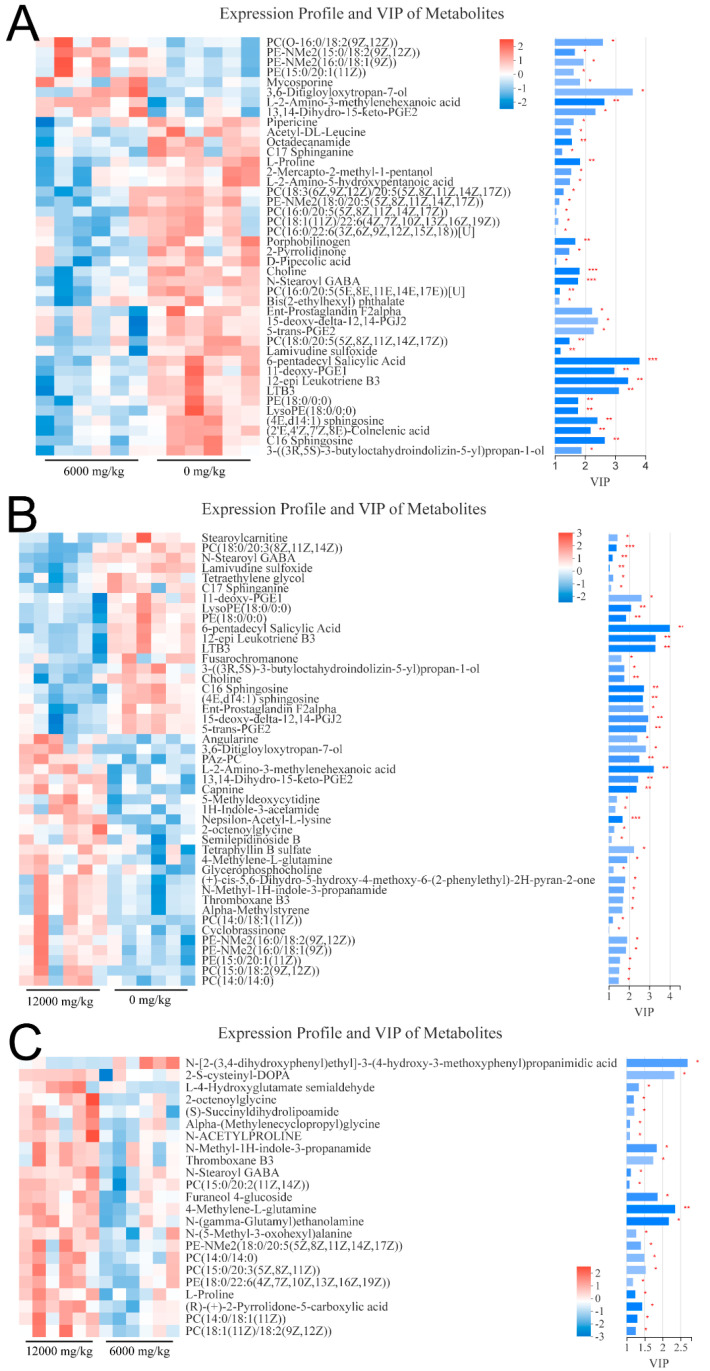
VIP values and expression levels of differential metabolites in the three compared groups: (**A**) 0 versus 6000 mg/kg; (**B**) 0 versus 12,000 mg/kg; (**C**) 6000 versus 12,000 mg/kg. * (*p* < 0.05), ** (*p* < 0.01) and *** (*p* < 0.001).

**Figure 3 animals-10-02246-f003:**
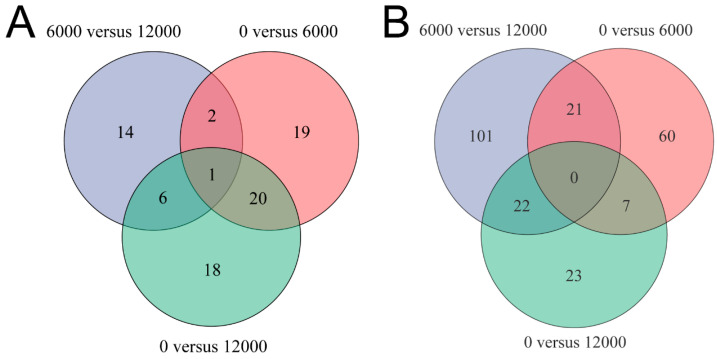
Venn diagram analysis among three compared groups: (**A**) differential metabolites; (**B**) differentially expressed genes.

**Figure 4 animals-10-02246-f004:**
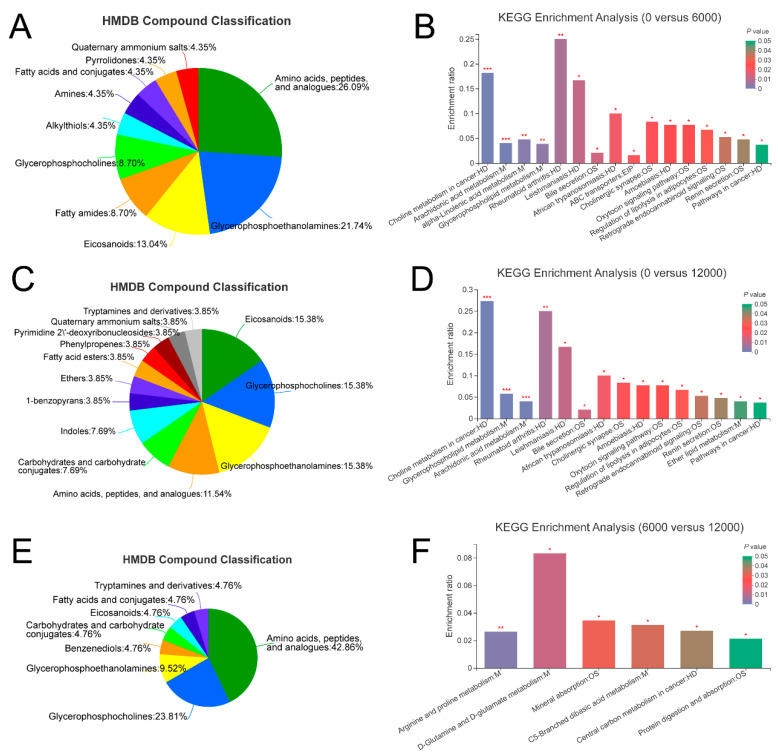
Classification of differential metabolites annotated to HMDB database in the three compared groups: (**A**) 0 versus 6000 mg/kg; (**C**) 0 versus 12,000 mg/kg; (**E**) 6000 versus 12,000 mg/kg. KEGG enrichment analysis based on differential metabolites in the three compared groups: (**B**) 0 versus 6000 mg/kg; (**D**) 0 versus 12,000 mg/kg; (**F**) 6000 versus 12,000 mg/kg. Levels of significance are defined as * (*p* < 0.05), ** (*p* < 0.01) and *** (*p* < 0.001).

**Figure 5 animals-10-02246-f005:**
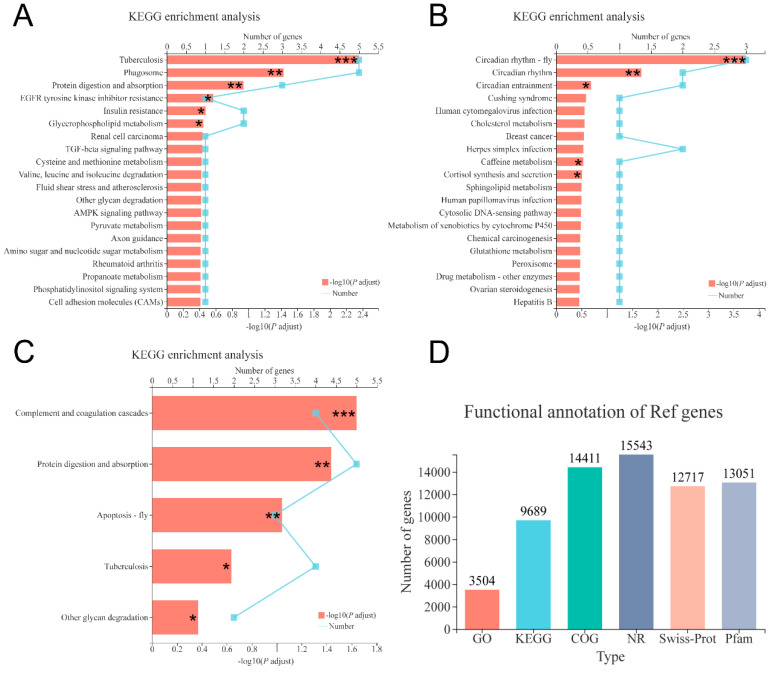
KEGG enrichment analysis based on DEGs in the three compared groups: (**A**) 0 versus 6000 mg/kg; (**B**) 0 versus 12,000 mg/kg; (**C**) 6000 versus 12,000 mg/kg. (**D**) Functional annotation information of genes obtained from six databases. Levels of significance are defined as * (*p* < 0.05), ** (*p* < 0.01) and *** (*p* < 0.001).

**Table 1 animals-10-02246-t001:** Ingredients and proximate composition of the basal diet.

Ingredients	g/kg of Dry Matter
Casein	400
Wheat flour	250
Gelatin	90
Fish oil	30
Soybean oil	30
Lecithin	10
Cholesterol	5
Choline-free vitamin premix ^1^	20
Mineral premix ^2^	20
Amino acid mixture ^3^	30
Butylated hydroxytoluene (BHT)	0.05
Alpha—cellulose	114.95
Choline chloride	0
Proximate composition (% of wet diet)	
Moisture	9.67
Crude protein	42.20
Crude lipid	7.99
Ash	2.04
Total choline (mg/kg diet)	1082

^1^ Vitamin premix (g/kg premix): vitamin A acetate (500,000 IU/g), 0.480; L-ascorbyl-2-polyphosphate 35% Active C, 35.710; folic acid, 0.180; biotin, 0.050; riboflavin, 3.000; DL Ca-pantothenate, 5.000; pyridoxine HCl, 1.000; vitamin B12, 0.002; thiamin HCl, 0.500; Menadione, 2.000; DL-alpha-tocopheryl acetate (250 IU/g), 8.000; inositol, 5.000; nicotinamide, 5.000; vitamin D (500,000 IU/g), 0.800; Bran flour, 933.278. ^2^ Mineral premix (g/kg premix): zinc sulfate monohydrate, 20.585; calcium iodate, 0.117; cupric sulfate pentahydrate, 0.625; manganous sulfate monohydrate, 1.625; magnesium sulfate monohydrate, 39.860; cobalt chloride, 0.010; ferrous sulfate monohydrate, 11.179; sodium selenite, 0.025; calcium hydrogen phosphate dihydrate, 166.442; bran flour, 759.532. ^3^ Amino acid mixture (g/kg of dry matter): glycine, 6; L-alanine, 6; L-glutamic acid, 6; betaine, 12.

**Table 2 animals-10-02246-t002:** Effects of different dietary choline chloride levels on the hepatosomatic index, weight gain, specific growth rate, condition factor, and survival of *L. vannamei* (n = 4, mean ± SE).

Choline Chloride Level (mg/kg Diet)	Hepatosomatic Index (%)	Weight Gain (%)	Specific Growth Rate (% day^−1^)	Condition Factor (%)	Survival (%)
0	3.89 ± 0.08	499.24 ± 20.52	3.18 ± 0.05	0.79 ± 0.01	73.33 ± 4.51
2000	3.97 ± 0.05	476.31 ± 22.97	3.09 ± 0.09	0.77 ± 0.01	78.33 ± 2.89
4000	3.86 ± 0.08	495.21 ± 14.38	3.13 ± 0.06	0.79 ± 0.00	77.50 ± 4.79
6000	3.81 ± 0.17	559.51 ± 21.48	3.35 ± 0.07	0.80 ± 0.01	73.33 ± 4.71
8000	3.76 ± 0.17	522.74 ± 24.39	3.27 ± 0.08	0.77 ± 0.01	76.67 ± 2.72
12,000	3.86 ± 0.20	491.60 ± 28.01	3.15 ± 0.10	0.78 ± 0.01	69.17 ± 3.44
*p* value	0.928	0.173	0.235	0.592	0.587

**Table 3 animals-10-02246-t003:** Proximate composition of *L. vannamei* fed different levels of dietary choline chloride (n = 4, mean ± SE).

Choline Chloride Level (mg/kg Diet)	Moisture (%)	Crude Protein (%)	Crude Lipid (%)	Ash (%)
0	76.38 ± 0.10	17.15 ± 0.20 ^ab^	2.04 ± 0.04	3.01 ± 0.03
2000	76.37 ± 0.40	17.39 ± 0.04 ^ab^	1.90 ± 0.10	3.14 ± 0.09
4000	76.40 ± 0.68	17.09 ± 0.31 ^ab^	1.78 ± 0.11	3.12 ± 0.02
6000	76.19 ± 0.29	17.52 ± 0.05 ^b^	2.01 ± 0.14	3.06 ± 0.04
8000	76.62 ± 0.20	16.73 ± 0.23 ^ab^	1.88 ± 0.06	3.01 ± 0.05
12,000	77.01 ± 0.13	16.61 ± 0.08 ^a^	1.85 ± 0.11	3.06 ± 0.10
*p* value	0.679	0.026	0.465	0.581

Values within the same column having different letters (a, b) indicate significant differences (*p* < 0.05).

**Table 4 animals-10-02246-t004:** Effects of different dietary choline chloride levels on the contents of total protein, glucose, triglyceride, and total cholesterol in the serum of *L. vannamei* (n = 4, mean ± SE).

Choline Chloride Level (mg/kg Diet)	Total Protein (g/L)	Glucose (mmol/L)	Triglyceride (mmol/L)	Total Cholesterol (mmol/L)
0	39.55 ± 2.63	2.27 ± 0.19	1.01 ± 0.12	2.09 ± 0.29 ^ab^
2000	40.51 ± 1.90	2.34 ± 0.13	1.16 ± 0.16	2.77 ± 0.38 ^ab^
4000	40.67 ± 1.72	2.26 ± 0.12	1.23 ± 0.17	2.15 ± 0.18 ^a^
6000	37.35 ± 1.77	2.16 ± 0.11	1.44 ± 0.06	3.40 ± 0.31 ^b^
8000	41.67 ± 1.77	1.95 ± 0.06	1.33 ± 0.06	3.64 ± 0.62 ^ab^
12,000	40.84 ± 2.15	2.00 ± 0.06	1.42 ± 0.15	3.76 ± 0.48 ^ab^
*p* value	0.741	0.075	0.095	0.007

Values within the same column having different letters (a, b) indicate significant differences (*p* < 0.05).

**Table 5 animals-10-02246-t005:** Effects of different dietary choline chloride levels on the total protein content, MDA content and activities of SOD, GSH-Px, AMS, and LPS in the hepatopancreas of *L. vannamei* (n = 4, mean ± SE).

Choline Chloride Level (mg/kg Diet)	Total Protein (g/L)	MDA (nmol/Mgprot)	SOD (U/Mgprot)	GSH-Px (U/Mgprot)	AMS (U/Mgprot)	LPS (U/Mgprot)
0	28.43 ± 1.69	4.40 ± 0.22 ^b^	2287.84 ± 38.51 ^b^	517.49 ± 89.61 ^b^	2.67 ± 0.28	0.41 ± 0.05
2000	38.31 ± 1.88	2.95 ± 0.16 ^a^	1560.96 ± 208.59 ^a^	380.54 ± 29.39 ^ab^	2.31 ± 0.22	0.27 ± 0.06
4000	39.05 ± 3.76	1.99 ± 0.24 ^a^	1345.69 ± 114.31 ^a^	214.99 ± 51.34 ^a^	2.56 ± 0.37	0.35 ± 0.07
6000	39.67 ± 4.11	2.95 ± 0.48 ^a^	1404.37 ± 87.10 ^a^	399.21 ± 59.51 ^ab^	2.45 ± 0.24	0.21 ± 0.05
8000	41.26 ± 3.81	2.84 ± 0.24 ^a^	1380.03 ± 88.90 ^a^	307.68 ± 34.89 ^ab^	2.26 ± 0.29	0.19 ± 0.04
12,000	39.60 ± 3.70	2.82 ± 0.43 ^a^	1612.94 ± 41.16 ^a^	409.22 ± 54.27 ^ab^	2.28 ± 0.37	0.19 ± 0.02
*p* value	0.103	0.000	0.000	0.016	0.910	0.076

Abbreviation: MDA, malondialdehyde; SOD, superoxide dismutase; GSH-Px, glutathione peroxidase; AMS, α-amylase; LPS, lipase. Values within the same column having different letters (a, b) indicate significant differences (*p* < 0.05).

**Table 6 animals-10-02246-t006:** Summary of the sequencing quality of *L. vannamei* given different levels of dietary choline chloride.

Choline Chloride Level (mg/kg Diet)	Clean Reads	Total Mapped (%)	Error Rate (%)	Q30 (%)	GC Content (%)
0	48,699,065	88.70	0.02	95.67	50.81
6000	51,050,008	88.41	0.02	95.70	50.93
12,000	51,946,736	88.12	0.02	95.71	50.54

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
