# Peer review of "Growth, Metabolite, Antioxidative Capacity, Transcriptome, and the Metabolome Response to Dietary Choline Chloride in Pacific White Shrimp Litopenaeus vannamei"

_animals, 2020, doi:10.3390/ani10122246_

Round 1
Reviewer 1 Report
The manuscript is well written in all aspects with only minor corrections needed in a few sentences in the Introduction and Discussion sections.
Ln 65: (delete the) from sentence. Besides choline...
Ln 350-351: in this study, no significant differences in.... were found (or were indicated) between different groups.
Ln 352: It was indicated that the total.....
Ln 368:..with the incremental increase of dietary choline..
Ln 420 (delete The) in the beginning of the sentence. P. monodon fed...
Author Response
The manuscript is well written in all aspects with only minor corrections needed in a few sentences in the Introduction and Discussion sections.
Ln 65: (delete the) from sentence. Besides choline...
Response: Thanks for your suggestion. We have revised the sentence as “Besides, choline also has many metabolic and physiological functions in aquatic animals.” in Line 69-70.
Ln 350-351: in this study, no significant differences in.... were found (or were indicated) between different groups.
Response: We have rewrote the sentence as “in this study, no significant differences in weight gain, survival, hepatosomatic index and condition factor were found between different groups.” in Line 335-336.
Ln 352: It was indicated that the total.....
Response: We have rewrote the sentence as “It indicated that the total choline level of 1082 mg/kg in the basic diet was sufficient for the growth needs of the L. vannamei.” in Line 336-337.
Ln 368:..with the incremental increase of dietary choline..
Response: Thanks for your suggestion. We have rewrote this paragraph.
Ln 420 (delete The) in the beginning of the sentence. P. monodon fed...
Response: We have deleted “the” in Line 383.
Reviewer 2 Report
Thanks for the high quality work submitted and well presented methodology in this work.
There are few points to be addressed by the authors:
1- show some of the LC-MS data is good to give the reader an idea about it.
2- The results represented in tables or figures contain few explanation in the captions, it is good to consider simplify it for the readers for easy following up.
Thanks
Author Response
reviewer 2
Thanks for the high quality work submitted and well presented methodology in this work.
There are few points to be addressed by the authors:
1- show some of the LC-MS data is good to give the reader an idea about it.
Response: In fact, we tried to show some differential metabolites in the table, but it made the article lengthy, and only a limited number of metabolites were discussed. All the differential metabolites and their VIP values are shown in Figure 2, and the details of differential metabolites are listed in the Supplementary Table S1-3.
2- The results represented in tables or figures contain few explanation in the captions, it is good to consider simplify it for the readers for easy following up.
Response: Thanks for your comment. We have simplified the captions.
Reviewer 3 Report
The manuscript reported the effect of choline in the Litopenaeus vannamei. The authors studied several parameters as growth performance, whole-body composition, serum characteristics, hepatopancreatic antioxidant indexes, serum metabolome and hepatopancreas transcriptome. The authors, in this work, obtained that choline diet decreased the oxidant damage in L. vannamei, however an excess choline can inhibit the digestion of protein. The work is quite interesting and the data obtained could be useful for the international scientific community. Also, the results obtained in this manuscript can be interest to aquiculture. The manuscript is well organized and well written. The article title clearly reflects the content of the manuscript, and the Abstract summarizes the content and the main results of the work. The methodology used was correctly described and the results were properly discussed.
For all these reasons, I suggest accept this paper.
Author Response
reviewer 3
The manuscript reported the effect of choline in the Litopenaeus vannamei. The authors studied several parameters as growth performance, whole-body composition, serum characteristics, hepatopancreatic antioxidant indexes, serum metabolome and hepatopancreas transcriptome. The authors, in this work, obtained that choline diet decreased the oxidant damage in L. vannamei, however an excess choline can inhibit the digestion of protein. The work is quite interesting and the data obtained could be useful for the international scientific community. Also, the results obtained in this manuscript can be interest to aquiculture. The manuscript is well organized and well written. The article title clearly reflects the content of the manuscript, and the Abstract summarizes the content and the main results of the work. The methodology used was correctly described and the results were properly discussed.
For all these reasons, I suggest accept this paper.
Response: Thanks for your approval.
Reviewer 4 Report
General comments
The present study evaluated the effects of choline levels on Lvannamei. A considerable effort was made to cover several aspects of these suplementaion levels. However, some points need to be improved in the manuscript. The introduction is adequate, with few suggestions. My biggest concern in this article is in the formulation of diets: some points need to be clarified and they are fundamental to guarantee the accuracy of the results and the conclusions. The models used to estimate the requirement do not seem adequate to me. The interpretation/description of some results is not supported by the tables presented. Consequently, part of the discussion needs to be revised. Please see the comments below:
Introduction
Some bibliographies cited studies on choline in the introduction used to fish as a biological model. Shrimp and fish have several physiological aspects different. My suggestion is to author focus on studies that has crustaceans as a biological model. Below, I list some studies not cited by authors:
Shi, B et al. 2020. Dietary choline improves growth performance, antioxidant ability and reduces lipid metabolites in practical diet for juvenile Pacific white shrimp,Litopenaeus vannamei. AQUACULTURE NUTRITION. DOI: 10.1111/anu.13163
Richard, L et al. The effect of choline and cystine on the utilisation of methionine for protein accretion, remethylation and trans-sulfuration in juvenile shrimp Penaeus monodon. BRITISH JOURNAL OF NUTRITION DOI: 10.1017/S0007114511001115
Michael, FR Effect of choline and methionine as methyl group donors on juvenile kuruma shrimp, Marsupenaeus japonicus Bate. AQUACULTURE DOI: 10.1016/j.aquaculture.2006.04.019
Lines 57-59: This phrase is confused. Please, rewrite. Is choline essential or not to M. japonicus (There is other comment about this below).
Lines 81-82: This sentence seems incomplete and unnecessary. Soon after, the authors describe the aim of the article. Please review this.
Material and Methods
Lines 98-101: Please provide more details on making diets. What proportion of water is added (%)? What is the extrusion temperature? It is an inappropriate and imprecise term. Correct it. Did choline replace the cellulose (1:1)? Clarify.
Table 1. Why you did not use a binder (e.g. carboxymethylcellulose) in the semi-purified diets? Gelatin and wheat flour are not enough to keep the stability of pellets underwater for a long time. Additionally, the high inclusion of cellulose (11%) decrease pellet stability. Shrimps manipulate diets a lot and eat slowly, and the choline is water-soluble... Perhaps this diet is adequate for fish studies, but I have doubt that is for nutritional studies with shrimp. Another aspect unclear in diets formulation: Why did you use whole casein and not a casein vitamin-free?
Tables 2, 3, 4 and 5: Please present P values for all variables. Please, show the Feed conversion ratio and other variables commented on previously in Table 2.
Line 154-158: Why feed conversion rate was not calculated? This is an essential parameter to be presented in the study. Why also were not calculate the productive value of protein, lipids, and energy? It is important to know the influence of choline levels on nutrient and energy retention.
Lines 230-239: What did was the criterion to choose the broken line two-slope model? Why not choose a broken line one-slope or other non-linear models? Please, clarify. Excel is software with few resources for regression analysis. In fact, I do not believe that the broken line two-slope model is suitable in these cases. For example, it is unjustifiable to propose a choline requirement of 3327 and 3769 for MDA and SOD, respectively, if with from 2000 there was no significant effect of choline levels on these variables. Why should I supply almost twice as much choline based on these models?
Results
Lines 246-248: This statement is not corroborated by the Tukey test in Table 3. In fact, the body crude protein of shrimps was different only between those fed with 6000 mg/kg (b) and 12000 mg/kg (a). The other treatments, including the control, showed intermediate values and were similar (ab) to the 6000 mg/kg (b) and 12000 mg/kg (a). This misinterpretation/description of the results also occurred for other variables.
Lines 266-267 – Again, this statement is not corroborated by Tukey’s test shown in Table 4. In fact, shrimp fed diets with 6000 mg/kg shown similar cholesterol levels (all with b letter) to fed other diets, except by those received 4000 mg/kg diets.
Lines 269-270 – If the statistical analysis was not significant, is inappropriate to describe an increase in serum triglycerides. In fact, all values are statistically equal. Please, correct it.
Lines 270-271 – What is the basis for describing enzyme activity as high? Other species? Other studies with L. vannamei? This sentence seems more suitable for discussion.
Line 277-278 - The previous comment on serum triglycerides applies here too when describing the result of total protein. Please, correct it.
Discussion
Lines 350-360: Most studies on choline in fish reported a requirement lower than 1,000 mg/kg. The control diet did have1082 mg/kg of "whole bound-choline" - most part from casein, probably. There is a previous study that reports a requirement of 600 mg/kg of choline for M. japonicus (NRC, 2011). Authors also reported a requirement for L. vannamei of 871 mg/kg (line 359). Thus, it is a controversial statement to write that choline is dispensable to M. japonicus or L. vannamei. I strongly recommend reviewing your approach to it. There was not “maximum weight gain” if the statistical analysis is no significant (line 356-358).
Lines 365-367 - Is really necessary to describe the results again?
Lines 375-376 - Why? The results do not support this statement. Weight gain was not affected in shrimp fed up to 12,000 mg / kg. From what value would the choline be in excess? The interpretation that shrimp fed 6000 mg/kg has deposited more protein than the others is wrong (see the previous comment). We must be careful when correlating gene expression data with performance data, for example. There are several unknown intermediate physiological mechanisms that prevent what is expressed genetically correspond to results registered.
Line 381 – Delete the first “damage”.
Line 400 - Perhaps cholesterol, triglycerides (P>0.05) definitely not.
Lines 401-405 - Just comparing results is not an adequate discussion.
Lines 409-433: I suggest the authors revise this paragraph. Most parts of the discussion showed is based on decreasing triglycerides in shrimp fed choline-deficient diets. However, this reduction not occurred truly because there was not a significant effect of choline levels on this parameter.
Lines 434-4448: I am not able to view the results of serum choline levels. Where are? Despite this, it is possible that control diet was not deficient. Suggest choline synthesis based only in serum choline levels is inadequate. I suggest a deep revision of this paragraph.
Line 457: Delete “were”.
Author Response
reviewer 4
General comments
The present study evaluated the effects of choline levels on L. vannamei. A considerable effort was made to cover several aspects of these suplementaion levels. However, some points need to be improved in the manuscript. The introduction is adequate, with few suggestions. My biggest concern in this article is in the formulation of diets: some points need to be clarified and they are fundamental to guarantee the accuracy of the results and the conclusions. The models used to estimate the requirement do not seem adequate to me. The interpretation/description of some results is not supported by the tables presented. Consequently, part of the discussion needs to be revised. Please see the comments below:
Introduction
Some bibliographies cited studies on choline in the introduction used to fish as a biological model. Shrimp and fish have several physiological aspects different. My suggestion is to author focus on studies that has crustaceans as a biological model. Below, I list some studies not cited by authors:
Shi, B et al. 2020. Dietary choline improves growth performance, antioxidant ability and reduces lipid metabolites in practical diet for juvenile Pacific white shrimp, Litopenaeus vannamei. AQUACULTURE NUTRITION. DOI: 10.1111/anu.13163
Richard, L et al. The effect of choline and cystine on the utilisation of methionine for protein accretion, remethylation and trans-sulfuration in juvenile shrimp Penaeus monodon. BRITISH JOURNAL OF NUTRITION DOI: 10.1017/S0007114511001115
Michael, FR Effect of choline and methionine as methyl group donors on juvenile kuruma shrimp, Marsupenaeus japonicus Bate. AQUACULTURE DOI: 10.1016/j.aquaculture.2006.04.019
Response: Thank you for your comments. The first and third paper has already quoted in [21] and [11], but the second paper was not quoted because it did not match the topic of our study.
Lines 57-59: This phrase is confused. Please, rewrite. Is choline essential or not to M. japonicus (There is other comment about this below).
Response: Thanks for your comment. This sentence was rewrote as “For example, the growth and survival of M. japonicus (about 0.004 and 0.01 g initial body weight, respectively) were reportedly improved by feeding on 600 and 1200 mg/kg dietary choline [6, 7]. However, Deshimaru and Kuroki (1979) had concluded that dietary choline chloride did not affect the growth of M. japonicus with 0.5 g initial body weight [8].” in Line 60-64.
Lines 81-82: This sentence seems incomplete and unnecessary. Soon after, the authors describe the aim of the article. Please review this.
Response: Thanks for your suggestion. We have deleted this sentence.
Material and Methods
Lines 98-101: Please provide more details on making diets. What proportion of water is added (%)? What is the extrusion temperature? It is an inappropriate and imprecise term. Correct it. Did choline replace the cellulose (1:1)? Clarify.
Response: This extruder does not have the function of temperature detection. We will improve it later. However, the addition of about 30% water reduces the pressure during extrusion (≤280 kg) and reduces the extrusion temperature. Moreover, the sentences were rewrote as “The choline chloride (reagent grade >98%; Sangon Biotech, Shanghai, China) was supplemented to the basal diet while reducing the same amount of α-cellulose. Finally, about 300 mL/kg distilled water was added to the mixture to produce a stiff dough, which was wet-extruded into 2.5-mm diameter pellets at an extrusion temperature of less than 50°C.(CD4-1TS extruder, SCUT industrial factory, China).” in Line 101-105.
Table 1. Why you did not use a binder (e.g. carboxymethylcellulose) in the semi-purified diets? Gelatin and wheat flour are not enough to keep the stability of pellets underwater for a long time. Additionally, the high inclusion of cellulose (11%) decrease pellet stability. Shrimps manipulate diets a lot and eat slowly, and the choline is water-soluble... Perhaps this diet is adequate for fish studies, but I have doubt that is for nutritional studies with shrimp. Another aspect unclear in diets formulation: Why did you use whole casein and not a casein vitamin-free?
Response: Question 1, we had conducted a pre-tests to confirm the stability of the diet after soaking in water and determined its suitability for Litopenaeus vannamei. In fact, some studies also did not use carboxymethyl cellulose (DOI: 10.1111/anu.13163; 10.1016/j.aquaculture.2018.09.056; 10.2331/suisan.45.363 and so on). Question 2, It is certainly best to use a vitamin-free casein. However, the choline requirement of L. vannamei is inconsistent. Some studies using whole casein show that the optimum choline chloride requirements of L. vannamei is 3254.1 - 6,488.3 mg/kg (DOI:10.7666/d.D604714; 10.1111/anu.13163), which is much more than the 1082mg/kg in our basial diet. Therefore, our basal diet can also serve as a choline deficiency diet. Thus, we think it is appropriate to use the whole casein.
Tables 2, 3, 4 and 5: Please present P values for all variables. Please, show the Feed conversion ratio and other variables commented on previously in Table 2.
Response: We have added the P values of these metrics to the tables. We did not calculate the feed conversion rate because we could not accurately measure the loss feed during the feeding experiment, and the feed conversion rate is not necessary to understand the physiological response of L. vannamei to dietary choline in this study.
Line 154-158: Why feed conversion rate was not calculated? This is an essential parameter to be presented in the study. Why also were not calculate the productive value of protein, lipids, and energy? It is important to know the influence of choline levels on nutrient and energy retention.
Response: We did not calculate the feed conversion rate because we could not accurately measure the loss feed during feeding. Of course, the productive value of protein, lipids and energy are important parameters for studying nutrition and energy retention. However, in this study, weight gain, whole body crude lipid and crude protein also played a role in evaluating the nutrient retention of L. vannamei. Moreover, we used metabolome and transcriptome to study the effects of choline on nutritional retention, energy metabolism and other physiological response of L. vannamei.
Lines 230-239: What did was the criterion to choose the broken line two-slope model? Why not choose a broken line one-slope or other non-linear models? Please, clarify. Excel is software with few resources for regression analysis. In fact, I do not believe that the broken line two-slope model is suitable in these cases. For example, it is unjustifiable to propose a choline requirement of 3327 and 3769 for MDA and SOD, respectively, if with from 2000 there was no significant effect of choline levels on these variables. Why should I supply almost twice as much choline based on these models?
Response: Thanks for your comment that help improve our article. We have deleted Figure 1.
Results
Lines 246-248: This statement is not corroborated by the Tukey test in Table 3. In fact, the body crude protein of shrimps was different only between those fed with 6000 mg/kg (b) and 12000 mg/kg (a). The other treatments, including the control, showed intermediate values and were similar (ab) to the 6000 mg/kg (b) and 12000 mg/kg (a). This misinterpretation / description of the results also occurred for other variables.
Response: We had simplified the description of the crude protein results.
Lines 266-267 – Again, this statement is not corroborated by Tukey’s test shown in Table 4. In fact, shrimp fed diets with 6000 mg/kg shown similar cholesterol levels (all with b letter) to fed other diets, except by those received 4000 mg/kg diets.
Response: We have rewrote the sentence as “In serum samples, the total cholesterol content in shrimp supplemented with 6000 mg/kg dietary choline was significantly higher than those of shrimp received 4000 mg/kg dietary choline (Table 4).” in Line 261-263.
Lines 269-270 – If the statistical analysis was not significant, is inappropriate to describe an increase in serum triglycerides. In fact, all values are statistically equal. Please, correct it.
Response: We have deleted this sentence.
Lines 270-271 – What is the basis for describing enzyme activity as high? Other species? Other studies with L. vannamei? This sentence seems more suitable for discussion.
Response: We have deleted this sentence.
Line 277-278 - The previous comment on serum triglycerides applies here too when describing the result of total protein. Please, correct it.
Response: We have deleted this sentence.
Discussion
Lines 350-360: Most studies on choline in fish reported a requirement lower than 1,000 mg/kg. The control diet did have 1082 mg/kg of "whole bound-choline" - most part from casein, probably. There is a previous study that reports a requirement of 600 mg/kg of choline for M. japonicus (NRC, 2011). Authors also reported a requirement for L. vannamei of 871 mg/kg (line 359). Thus, it is a controversial statement to write that choline is dispensable to M. japonicus or L. vannamei. I strongly recommend reviewing your approach to it. There was not “maximum weight gain” if the statistical analysis is no significant (line 356-358).
Response: We have rewrote this paragraph.
Lines 365-367 - Is really necessary to describe the results again?
Response: We have deleted this sentence and rearranged the paragraph.
Lines 375-376 - Why? The results do not support this statement. Weight gain was not affected in shrimp fed up to 12,000 mg / kg. From what value would the choline be in excess? The interpretation that shrimp fed 6000 mg/kg has deposited more protein than the others is wrong (see the previous comment). We must be careful when correlating gene expression data with performance data, for example. There are several unknown intermediate physiological mechanisms that prevent what is expressed genetically correspond to results registered.
Response: We have rewrote this paragraph.
Line 381 – Delete the first “damage”.
Response: We have deleted the first “damage”.
Line 400 - Perhaps cholesterol, triglycerides (P>0.05) definitely not.
Response: We have corrected it.
Lines 401-405 - Just comparing results is not an adequate discussion.
Response: We have rewrote this paragraph.
Lines 409-433: I suggest the authors revise this paragraph. Most parts of the discussion showed is based on decreasing triglycerides in shrimp fed choline-deficient diets. However, this reduction not occurred truly because there was not a significant effect of choline levels on this parameter.
Response: We have rewrote this paragraph and deleted the discussion about triglycerides.
Lines 434-4448: I am not able to view the results of serum choline levels. Where are? Despite this, it is possible that control diet was not deficient. Suggest choline synthesis based only in serum choline levels is inadequate. I suggest a deep revision of this paragraph.
Response: The results of serum choline were listed in Supplementary Table S1 and S2. We have rewrote this paragraph.
Line 457: Delete “were”
Response: We have deleted “were”.
Round 2
Reviewer 4 Report
Congratulations on your research.